# Athletics—The World’s Most Measurable Sport, but at What Price? A Comparison of Fully Automatic Timing with Times Measured with Standard Mobile Phones

**DOI:** 10.3390/s22072774

**Published:** 2022-04-04

**Authors:** Bjørn Johansen, Roland van den Tillaar

**Affiliations:** Department of Sports Sciences, Nord University, 7600 Levanger, Norway; bjorn.johansen2@student.nord.no

**Keywords:** timing, iPhone, athletics, video camera, sprint running

## Abstract

The aim of this study is to quantify potential systemic timing bias between fully automatic timing (FAT) and timing with iPhone camera (Apple Inc., Cupertino, CA, USA) and then consider whether an iPhone can be used as an inexpensive timing system for sprint events at athletics competitions. A flashlight was aimed at FAT camera (Lynx System Developers, Haverhill, MA, USA) and two iPhones, at 120 and 240 frames per second (fps), respectively, so that they could capture the light from it. By turning the flashlight on and off at varying intervals (1–33 s, average 9.5 s), the cameras captured a series of light beams. The time intervals between the start of two light beams were measured 31 times on the recordings from all the cameras. On each recording with the iPhones, two analyses were performed: one where the video image before the light beam (start before light) from the flashlight was set to 0 s and one where the first image with the light beam (start on light) was set to 0 s. Start on light showed no significant time differences compared to FAT. With 240 fps the standard deviation was ± 0.001 s, 29% of the times were the same as FAT, while 81% of the times are within ± 0.001 s. The largest deviation was a time of −0.003 s from FAT. With 120 fps there was a standard deviation of ± 0.003 and a maximum deviation of −0.006 s, where 39% of the times were within ± 0.001 s. At start before light, a significant but expected difference was found with an average deviation of +0.008 s with 120 fps and +0.004 s with 240 fps, with maximum deviations of +0.014 and +0.006 s. It can be concluded that the camera on an iPhone is accurate as we did not find any systematic bias from FAT with start on light.

## 1. Introduction

In athletics, times in the running events must be measured accurately, especially in sprint distances, where the difference between runners often is within a few hundredths of a second. The international governing body for the sport of athletics, World Athletics (WA) requires fully electronic timing (FAT) that have been tested and have a certificate of accuracy at a minimum to the hundredth of a second for sprint events [1]. The timing system is started automatically by a gun with a maximum delay of less than 0.001 s and producing a composite image with a line-scan camera which takes very narrow vertical images of the finish line. However, such timing equipment is expensive, impractical, and it is time consuming for small-meet organizers to set up. The question arises if we do need so expensive, accurate, and advanced equipment for smaller meets, when in other events in athletics only measuring tapes in cm are required for measurement in jumps and throws?

Nowadays video cameras in mobile phones are capable of recording 100 frames per second (fps) or higher (which is in 100th of a seccond), which is similar to acquirements of accuracy in sprint times. Since the release of the iPhone 6, the iPhones’ high-speed camera has been able to film at 120 and 240 fps. In theory, analyzing the recording of the smoke or flame from the starter gun and sprinters passing the finish line with a video camera, should provide enough information for valid sprint time. With a frame rate of 240 fps, you can in theory get a maximum error of about 0.004 s, which is less than a hundredth of a second error, as sprint times is measure in. Haugen, et al. [2] compared FAT system for athletics competition with recordings from a video camera and found an absolute variation of 0.01 s but no mean or systematic variation. Recent innovation in smartphone technology has led to the development of several new applications for measuring physical performance. SpeedClock and MySprint are apps that use mobile phone camera and measure the speed of an object. When these apps were compared to single beam photocells, there was an almost perfect correlation with r = 0.93 and r = 0.989–0.999, respectively [3,4]. But despite this, there is a potential source of error when using single beam photocells to measure time. Different body parts can activate the cells and Haugen, et al. [5] found a time difference in the range of ±0.05–0.06 s between single and double photocells.

In addition to these, there are many apps that measure time and speed of athletes and objects that have not been scientifically analyzed [6,7,8]. These apps use the iPhone camera and support accurately measure times in 100 and 1000 part of a second. To the best of our knowledge, there are currently no studies that demonstrate the validity of an iPhone as a timing system compared to FAT. Therefore, the aim of this study is to quantify potential systemic timing bias between fully automatic timing and timing with iPhone camera and then consider whether an iPhone can be used as an inexpensive timing system for sprint events at athletics competitions. 

## 2. Materials and Methods

### 2.1. Procedure

To compare the automatic timing device with a regular iPhone camera, a FinishLynx FAT camera (Lynx System Developers, Haverhill, MA, USA) and two iPhones, one iPhone 8 (Apple Inc., Cupertino, CA, USA) operating at 120 fps and one iPhone 6 operating at 240 fps were used. FinishLynx photo finish timing is a fully automatic timing system that was used as the “gold standard” for the validation of timing with iPhone camera, as this system is used in many competitions and approved by the WA for timing at all levels of competitions [9]. In competitions, the system starts by firing a gun, and the sound is captured by a sensor. The camera is directly aligned with the finish line and captures 1000 images per second. FinishLynx is a line-scan camera that takes very narrow vertical images of the finish line. These images are then placed besides each other and composite images of the athletes. Each image corresponds to a time, providing information to read the time of the athlete.

According to the competition rules of WA [1], the athletes time is taken when the torso reaches the vertical plane of the nearer edge of the finish line. Because the aim of this study was to quantify potential systemic timing bias between FAT and iPhone, it was important to get an exact reading of both the FAT picture and the video images from the iPhone. To read a picture of a running human can be problematic since the foremost part of the torso can be difficult to define when the shoulder and the arm is in front of the chest. To eliminate this potential problem, a flashlight was aimed at the various cameras so they could capture the light from it. All three cameras were located about 10 cm from the flashlight, where the FAT camera was aimed directly at the light source, and the iPhones were placed to the side of the source. The FAT system was started with a pat of the hands close to the sensor, while the iPhones were started manually by pressing the on button on the camera (Figure 1). By turning the flashlight on and off at varying intervals (1–33 s, average 9.5 s), the cameras captured a series of beams of light.

The time intervals between the start of two light beams were measured 31 times on the recordings from all the cameras. On each recording with the iPhones, two analyses were performed: one in which the video image before the light beam (start before light) from the flashlight was set at 0 s and one in which the first image with the light beam (start on light) was set at 0 s. The images from the FAT camera were analyzed using FinishLynx software (Lynx System Developers, Haverhill, MA, USA), while the video recordings obtained by the iPhones were analyzed in Kinovea.0.8.15. The software identified the time by entering frames per second, and identified the image before the light/first image with light as the start and first image with light on the next light beam as the end time. Comparison between the different timing systems was determined based on mean and standard deviation from FAT. 

### 2.2. Statistical Analysis

To compare the results of the timing devices a paired samples T-test was used to identify eventual significant differences between FAT and iPhone timing. Significance was accepted at the *p* ≤ 0.05 level. All times are reported to the nearest 0.001 s. Deviation from FAT is also presented with figures as a frequency distribution.

## 3. Results

A significant difference in timing between the two systems was found when start of timing started on the image before light beam. The total times were on average 0.008 and 0.003 s longer when measured with resp. 120 and 240 fps (Figure 2). No significant differences were found when timing started on first image with light beam (Table 1).

## 4. Discussion

This study compared a fully automatic timing system with the camera on an iPhone to investigate if a mobile phone can be used as an accurate timing system. When timing started on the first image with light beam, the analysis revealed no systematic bias or significant differences between FinishLynx fully automatic timing system and video timing with iPhone 6 and iPhone 8.

With start on light and 120 fps you can in theory expect an average and maximum deviation of 0.000 and ± 0.008 s, while with start on light and 240 fps you can expect an average and maximum deviation of 0.000 and ± 0.004 s. Both deviations are within 0.01 s in which the times at athletics competitions are measured. In this study the videos with 240 fps and start on light showed small deviations from FAT. The standard deviation is ±0.001 s, 29% of the times were at exactly the same thousandth of a second as FAT, while 81% of the times are within ± 0.001 s. The largest deviation was a time of −0.003 s from FAT. There were also relatively small deviations with start on light and 120 fps with a standard deviation of ± 0.003 and a maximum deviation of −0.006 s where 39% of the times were within ± 0.001 s.

With start before light and 120 fps you can in theory expect an average and a maximum deviation of +0.008 and +0.016 s. With start before light and 240 fps, you can in theory expect an average and a maximum deviation of +0.004 and +0.008 s. The figures from this study agree with the theoretical figures with an average deviation of +0.008 s with 120 fps and +0.004 s with 240 fps, with maximum deviations of +0.014 and +0.006 s (Figure 2). To the best of our knowledge, no one has tested video timing with iPhone camera against FAT before. FinishLynx is a tested and approved FAT system that is certified by WA [9]. Haugen, Tønnessen, and Seiler [2] compared the recording with 50 fps from video camera with FAT. To calculate the time from the video analysis in 0.01 s, they looked at the size of the plume of smoke from the gun to start the time and where the runner was in relation to the finish line to stop the time. Based on this, they found a precision of ± 0.01 s. 

WA has rules for rounding off times that can have a major impact on the result. The rules say that all times of track races up to 10,000 m should be recorded to a precision of 0.01 s or the time should be converted and recorded to the next longer 0.01 s, e.g., 10.001 s must be registered as 10.01 s [1]. This means that you can get a deviation of 0.009 s from the official results, even if the real deviation was only 0.001 s. At 31 registered times with the iPhones with start on light at 240 fps, there were four times that deviated by ± 0.01 s on the official results. At 31 registered times with start before light with 240 fps, there were 15 times that deviated by +0.01 s on the official result, that is, about half the times. So even if the real time deviates by less than a hundredth of a second, you can get a deviation of 0.01 on the official result. In practice, this should not be something that excludes timing with the iPhone, when there is such a small deviation. But such a deviation can have unfair consequences when it comes to ranking and qualifying for e.g., championships etc. Before digital line-scan cameras were introduced, there was a period WA approved video timing [10]. The finish line was filmed with a camera that took at least 50 fps. With only 50 fps, you could only take every other hundredth of a second and unless the time was exactly 0.01 s, the time should be converted and registered to the next higher 0.01 s, e.g., 10.001 must be registered as 10.02. By using start before light and 240 fps, you will always be able to get an official time that is equal to or 0.01 more than FAT. Then you do not risk those times with the iPhone which will be beneficial in relation to FAT. But if you made an app and a sensor that captures the sound from the shot and automatically starts a clock in the camera, then there will only be times equal or longer than FAT.

A limitation in this present study should be addressed. This study has not been conducted during a regular competition, but under controlled conditions with light beams. So, it cannot be directly transferred to a competitive situation. Therefore, one should do studies under regular competition conditions where one records the gun shot and sprinters passing the finish line and comparing the times from the iPhones with the official results. 

Based upon the findings of this study it can be concluded that the camera in an iPhone is accurate as we did not find any systematic bias from FAT with start on light. While the systematic bias with start before light was exactly within what one could theoretically expect. 

The iPhones show accurate timing and will be a good alternative in athletics competitions as it is cheaper and easier to set up and move (when changing places due to wind). In practice, the camera on the iPhone must detect the flame/smoke from the gun and when the runner crosses the finish line and then analyze the total time from smoke to finish line in a computer program in which videos can be loaded with correct frames per second. With a FAT camera you also need a front camera to identify the race number and the track the runner is in. But with a video camera, you only need one camera, since you can see their race number and which lane they are in before they pass the finish. Furthermore, it can also be used in the laboratory instead of single beam photocells as these have an accuracy of ± 0.06 s [5]. You only need to register the start signal and then the camera can measure the finish. 

## Figures and Tables

**Figure 1 sensors-22-02774-f001:**
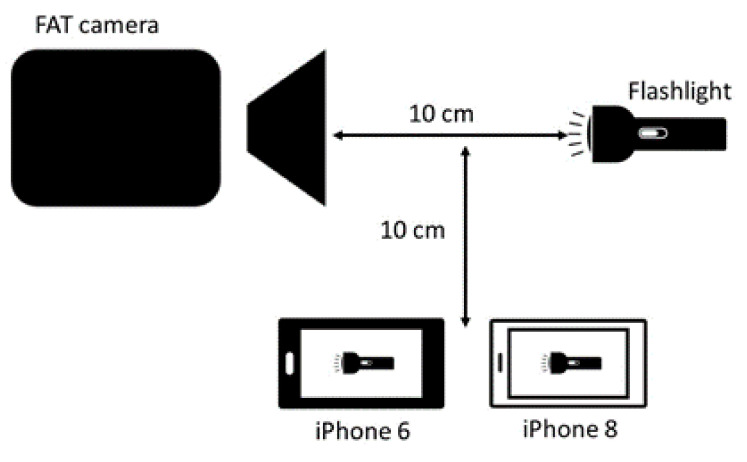
Set up with FAT camera, iPhones, and light source.

**Figure 2 sensors-22-02774-f002:**
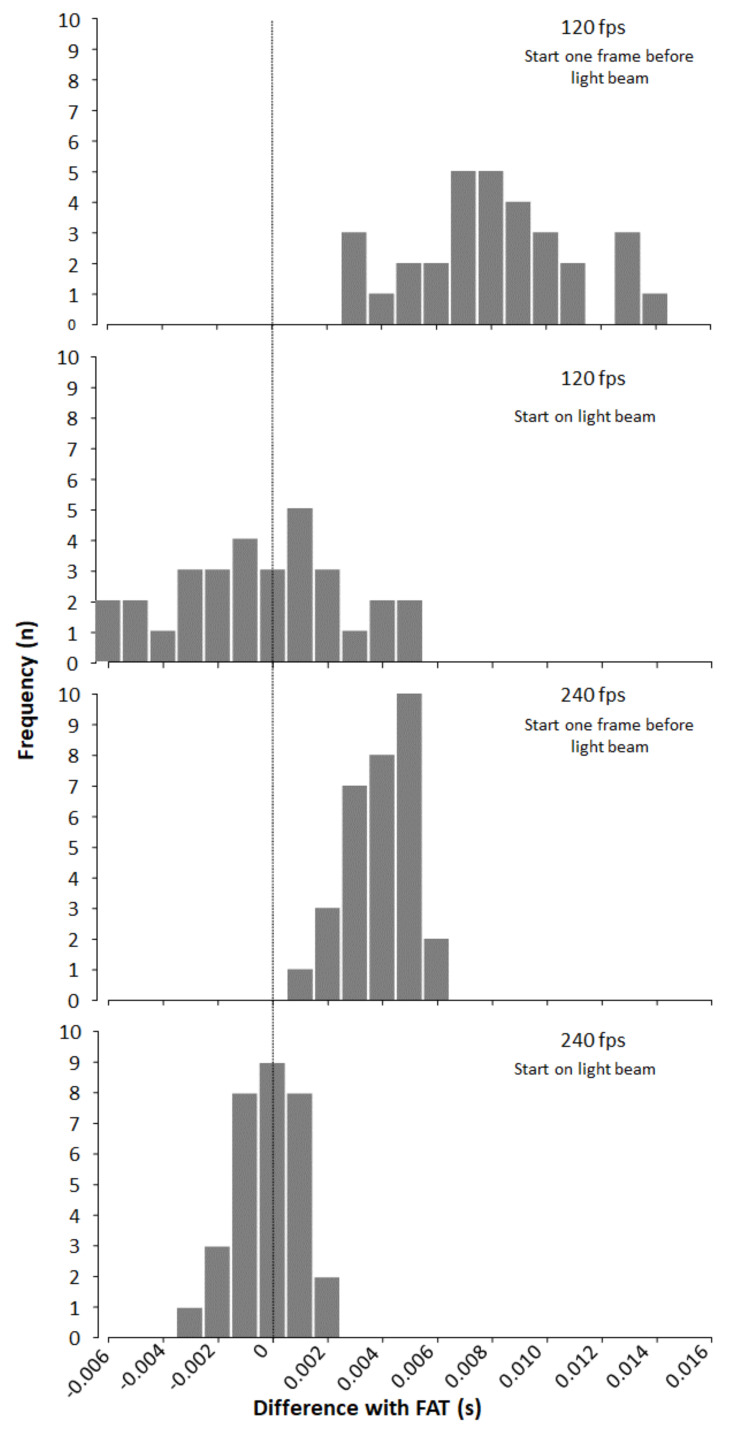
Difference between iPhones and the FAT.

**Table 1 sensors-22-02774-t001:** Main difference (±SD) between iPhones and the FAT.

Fps and Start Image	n	Δ (s)
iPhone 120 fps start timing on image before light beam	31	0.008 ± 0.003 *
iPhone 120 fps start timing on first image with light beam	31	0.000 ± 0.003
iPhone 240 fps start timing on image before light beam	31	0.004 ± 0.001 *
iPhone 240 fps start timing on first image with light beam	31	0.000 ± 0.001

* indicates a significant difference on a *p* < 0.05 level.

## Data Availability

The data presented in this study are available on request from the corresponding author. The data are not publicly available due to national laws of the Norwegian government on privacy.

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
