# Peer review of "Athletics—The World’s Most Measurable Sport, but at What Price? A Comparison of Fully Automatic Timing with Times Measured with Standard Mobile Phones"

_sensors, 2022, doi:10.3390/s22072774_

Round 1
Reviewer 1 Report
The paper makes a potentially worthy contribution to knowledge, in that validation of technologies is a major barrier to adoption. Having said that this paper takes a blackbox approach to the technology investigated. This might be ok for a sport science journal, but for a journal focused on sensors and technology it needs to go deeper. Further, consideration towards implimentation and practicality needs to be also addressed to establish the utility of the methods discussed.
Some suggestions to improve theaccuracy and readabilityof the paper
- Consider working in milliseconds perhaps and a uniform adoption of units across the paper.
- Clearly articulate a frame number (relative or absolute) so that the various video sources (and frame rates) can be clearly demonstrated to the reader, where multiple sources (sample rates and fps) are used.
- Claims of accuracy of 0.01s in a historical reference (Haugen et al.) would seem to be inconsistent with a 50fps (this is 0.02s ?)
- Clearly articulate how the timing information is recovered from the iPhones - iOS is this through counting frames? How would this work in a practical sense?
- The reference for sample rate limitations of photoreceptors is over 10 years old is this consistent with the technologies used
- Timing differences with iPhones (start on or before) are indicative of systems issues with the iPhone operating system. in your considerations and discussion try to take the reader deeper here. The work of Rowlands et. al and iPhone systems might be useful reading for insight.
- Figure 2 is misleading. If the X axis is in fps why does the data between 120 and 240 fps have the same number of datum across the sub plots? (See points 1 & 2)
Author Response
Thanks for reviewing the manuscript and the helpful feedback. We have written our answers to them point by point below. We think that it is now suitable for publication.
The paper makes a potentially worthy contribution to knowledge, in that validation of technologies is a major barrier to adoption. Having said that this paper takes a blackbox approach to the technology investigated. This might be ok for a sport science journal, but for a journal focused on sensors and technology it needs to go deeper. Further, consideration towards implementation and practicality needs to be also addressed to establish the utility of the methods discussed.
Some suggestions to improve the accuracy and readability of the paper
- Consider working in milliseconds perhaps and a uniform adoption of units across the paper.
It was possible to use milliseconds. However, in scientific articles it is normal to use the SI units. Furthermore, in athletics it is always used in seconds. Hope it's okay.
- Clearly articulate a frame number (relative or absolute) so that the various video sources (and frame rates) can be clearly demonstrated to the reader, where multiple sources (sample rates and fps) are used.
As we have written before iPhone and FAT have different frame rates (120/240 fps with iPhones and 1000 fps with FAT). In athletics differences are measured in seconds, not in fps. The main purpose was to investigate the timing differences between the two systems, as that is what matters if you want to use the iPhone camera as a timing system and thereby use seconds and not compare it in fps.
- Claims of accuracy of 0.01s in a historical reference (Haugen et al.) would seem to be inconsistent with a 50fps (this is 0.02s ?)
We understand that this may be unclear as Haugen et al. had only 50 fps. They "calculated" the time in 0.01 sec by looking at the size of the smoke cloud from the gun and where the athlete is in relation to the finish line when they take the finish time. I will elaborate on this in the article now.
- Clearly articulate how the timing information is recovered from the iPhones - iOS is this through counting frames? How would this work in a practical sense?
Kinovea software automatically calculates the time when you enter frames per second in the application. I have elaborated on this a bit in the article now.
- The reference for sample rate limitations of photoreceptors is over 10 years old is this consistent with the technologies used
The technology has not evolved significantly since then, so we think this is a relevant reference.
- Timing differences with iPhones (start on or before) are indicative of systems issues with the iPhone operating system. in your considerations and discussion try to take the reader deeper here. The work of Rowlands et. al and iPhone systems might be useful reading for insight.
In the "theory" there should "almost" always be 0.004 sec worse time with the picture before the light, so then you really only need to add 0.004 sec to the time with the first picture with the light. But since there is not exactly 0.004 sec between each image, but 0.0041666 .. (1 sec / 240 fps) it is sometimes 0.005 sec between two images to "correct" this small discrepancy. This is why I have a own "category" with image before light.
- Figure 2 is misleading. If the X axis is in fps why does the data between 120 and 240 fps have the same number of datum across the sub plots? (See points 1 & 2)
We understand that it may be somewhat unclear, but the reason is that we focus on the time difference and not the difference in fps since that is what has been done in previous studies and it is the time that is measured with FAT.
Reviewer 2 Report
Dear authors, this is a topic of interest to practitioners. I have one minor comment and one major comment.
Introduction - The background could be more substantial. What do we know from studies that use timing gates for example?
The study had two aims. One was to quantify potential systematic timing bias between fully auto-matic timing (FAT) and timing with an iPhone camera, and that comparison was achieved.
The second aim stated was to consider whether an iPhone can be used as an inexpensive timing system for sprint events in athletic competitions. It is not clear whether this was achieved.
As correctly identified by the authors, "reading a picture of a running human can be problematic...". What then is the proposed solution in order to utilise the more affordable and accessible option of a smart phone for timekeeping? How will the study results be applied in actual sporting context?
Author Response
Dear authors, this is a topic of interest to practitioners. I have one minor comment and one major comment.
Thanks for the nice answer and helpful feedback. We have responded to your comments below.
Introduction - The background could be more substantial. What do we know from studies that use timing gates for example?
Introduction - We think that since time gates cannot be used in competitions and are not as accurate as the light beam can be broken by different body parts from trials to trials it is not so interesting to elaborate more for this study. Furthermore, it can also only measure one person at a time. This study wanted to look at a more accurate system down to the 1000th of a second that can measure multiple athletes at a time.
The study had two aims. One was to quantify potential systematic timing bias between fully auto-matic timing (FAT) and timing with an iPhone camera, and that comparison was achieved.
The second aim stated was to consider whether an iPhone can be used as an inexpensive timing system for sprint events in athletic competitions. It is not clear whether this was achieved.
As correctly identified by the authors, "reading a picture of a running human can be problematic...". What then is the proposed solution in order to utilise the more affordable and accessible option of a smart phone for timekeeping? How will the study results be applied in actual sporting context?
We have explained in a little more detail how the camera on an iPhone can be used practically in competitions to measure the time of runners, so this becomes a little clearer to the reader.